# Poor outcomes among critically ill HIV-positive patients at hospital discharge and post-discharge in Guinea, Conakry: A retrospective cohort study

**Sebastian Ludwig Albus**[1‡]*, **Rebecca E. Harrison**[1‡], **Ramzia Moudachirou**[1],
**Kassi Nanan-N'Zeth**[1], **Benoit Haba**[1], **Esther C. Casas**[2], **Petros Isaakidis**[2],
**Abdourahimi Diallo**[1], **Issiaga Camara**[3], **Marie Doumbuya**[3], **Fode Bangaly Sako**[3],
**Mohammed Cisse**[4]

1 Medecins Sans Frontieres, Guinea Mission, Operational Centre Bruxelles, Bruxelles, Belgium, 2 Medecins Sans Frontieres, Southern African Medical Unit, Cape Town, South Africa, 3 Gamal Abdel Nasser University of Conakry Faculty of Medicine, Unite de Soins, Formation et de la Recherche (USFR), Conakry, Guinea, 4 Gamal Abdel Nasser University of Conakry Faculty of Medicine Pharmacy and Odonto- Stomatology, Conakry, Guinea

‡ SLA and REH are contributed equally to this work and considered joint first authors.
* seb.albus@gmail.com

**Data Availability Statement:** All relevant data are within the paper and its Supporting information files.

## Abstract

### Introduction

Optimal management of critically ill HIV-positive patients during hospitalization and after discharge is not fully understood. This study describes patient characteristics and outcomes of critically ill HIV-positive patients hospitalized in Conakry, Guinea between August 2017 and April 2018 at discharge and 6 months post-discharge.

### Methods

We carried out a retrospective observational cohort study using routine clinical data. Analytic statistics were used to describe characteristics and outcomes.

### Results

401 patients were hospitalized during the study period, 230 (57%) were female, median age was 36 (IQR: 28–45). At admission, 229 patients (57%) were on ART, median CD4 was 64 cells/mm³, 166 (41%) had a VL >1000 copies/ml, and 97 (24%) had interrupted treatment. 143 (36%) patients died during hospitalisation. Tuberculosis was the major cause of death for 102 (71%) patients. Of 194 patients that were followed after hospitalization a further 57 (29%) were lost-to-follow-up (LTFU) and 35 (18%) died, 31 (89%) of which had a TB diagnosis. Of all patients who survived a first hospitalisation, 194 (46%) were re-hospitalised at least once more. Amongst those LTFU, 34 (59%) occurred immediately after hospital discharge.

**Funding:** This work was supported through routine program costs by Médecins Sans Frontiers, Operational Centre Belgium.

**Competing interests:** None of the authors report any competing interests or conflict of interests.

**Abbreviations:** ART, Anti-retroviral treatment; CNERS, Comité National d'Ethique et Recherche; CS, centre de santé; GFR, Glomerular Filtration Rate; HAART, Highly Active Anti Retro Viral Therapy; HIV, Human Immunodeficiency virus; IPD, In patient department; IQR, Interquartile Range; LAM, Lipoarabinomannan; LLV, Low level Viraemia; MOH, Ministry of Health; MSF, Medecins Sans Frontieres; MUAC, middle upper arm circumference; NNRTI, Non nucleosidic reverse transcriptase inhibitor; NRTI, Nucleosidic reverse transcriptase inhibitor; OI, Opportunistic Infection; OPD, Out-patient department; PCP, Pneumocystis carinii pneumonia; PI, Protease inhibitor; TB, Tuberculosis; TBM, Tuberculosis Meningitis; USFR, Unité des Soins, Formation et Recherche; VL, Viral Load; WCA, West and Central Africa.

## Conclusion

Outcomes for critically ill HIV-positive patients in our cohort were poor. We estimate that 1-in-3 patients remained alive and in care 6 months after their hospital admission. This study shows the burden of disease on a contemporary cohort of patients with advanced HIV in a low prevalence, resource limited setting and identifies multiple challenges in their care both during hospitalisation as well as during and after re-transitioning to ambulatory care.

## Introduction

While roll out of Anti-Retroviral Treatment (ART) in Sub Saharan Africa has decreased, the burden of HIV [1] and mortality due to HIV remains elevated in Western and Central Africa (WCA) [2]. Advanced HIV disease remains a major issue, both due to late presentation of new patients as well as ART-experienced patients who drop out of treatment [3]. Availability and quality of HIV care differs between the low prevalence countries of Western and Central Africa (WCA) and Southern and Eastern Africa (SEA) [4]. Despite only contributing 17% of world-wide HIV cases, WCA accounts for 31% of HIV mortality globally, as well as 21% of new infections [4]. In Guinea, about 120,000 people live with HIV, the prevalence within the adult population (15–49 years) is low (1.5%), [5] and around 35% of HIV positive adults are estimated to be on ART [5, 6]. Access to diagnostic testing for opportunistic infection (OIs), HIV viral load or CD4 count remain limited and HIV programs encounter intermittent drug supply ruptures.

Recent improvements in the management of advanced HIV-patients include the provision of tailored treatment interventions and the recommendation to consider switch from $1^{st}$ to $2^{nd}$ line ART on only 1 elevated VL in severely ill patients with a fitting history [7, 8], although it is currently unclear whether these new recommendations are adhered to in practice. Acquired drug resistance in patients who fail $1^{st}$ line treatment is of growing concern [8], with a Guinean study from 2014 among treatment experienced patients with virologic failure reporting at least one drug resistance mutation (DRM) in 68% of cases [9]. Recent WHO data for neighbouring Senegal is similar, where 7% of all patients on ART and 63% of patients with a viral load >1000copies/ml showed at least one DRM [10]. Also, failing patients in Sub-Saharan Africa still present with low CD4 counts, making effective treatment and rapid immune reconstitution a priority [7, 11]. Primary resistance is rare, with one small Guinean study estimating the prevalence of DRM in treatment naïve patients as low as 6.45% [12, 13].

MSF has supported HIV care in Guinea since 2003. Since 2016, MSF supports the unit of care, training and research (USFR), a tertiary level referral hospital run by the ministry of health (MOH). USFR is dedicated to the care of HIV patients in need of hospitalisation. Until now however, there is virtually no data on the long term outcomes of patients with advanced HIV in the WCA region. This study aims to address this gap by describing profiles and outcomes at discharge and post-discharge of critically ill HIV-positive patients who were hospitalised at USFR, Conakry, Guinea, between August 2017 and April 2018.

## Methods

### Study design

We carried out a retrospective observational cohort study on routine data of all patients with HIV infection admitted to USFR Donka between August 2017 and April 2018. Post-discharge

from hospital, patients returned to ambulatory care, and we documented 6-months follow up post-discharge.

**Study setting.** HIV care in Guinea is provided at primary health care level where patients are followed in integrated MOH outpatient departments (OPD). Admission criteria to USFR are algorithmized and based on clinical severity (e.g. positive shock index, GCS<12 etc.). USFR receives patients as direct referrals from seven health centers located in different parts of Conakry (CS Matam, CS Coleah, CS Wanindara, CS Gbessia port, CS Flamboyant, CS Coleah, CS Tombolia). These health centers together serve a large OPD cohort of around 10,000 HIV patients. The main source of patients' income is informal vending, the rate of food insecurity is high and HIV associated stigma is widespread. ART treatment in Guinea is free, with 1st line treatment usually consisting of two nucleosidic reverse transcriptase inhibtors (NRTI) plus one non nucleosidic reverse transcriptase inhibitor (NNRTI). 2nd line treatment consists of a two NRTI plus one boostered Protease Inhibitor (PI).

USFR has a total of 31 beds including 4 intensive care beds. At USFR, patients receive lab tests, drugs, and treatment free of charge. Admission algorithms include routine testing of CD4 count, HIV VL, full blood count, biochemistry, Serum CrAg, Malaria rapid test and TB investigations with Urine LFA TB-LAM and GeneXpert testing on sputum or gastric lavage. CSF analysis, Sputum micoscropy and X-Ray are available off site.

Post-discharge, cases are routinely referred to CS Matam, a specialized secondary level HIV-OPD. Once stabilised further, they are referred to their original OPD.

During the study period, treatment failure at USFR was defined in accordance with the 2016 WHO recommendations in place at the beginning of the study period (2 VL>1000, 3 months apart despite adherence counselling) [14], but patients could be switched on only one elevated VL or at the attending clinician's discretion. Of note, we chose a middle upper arm circumference (MUAC) of <210mm instead of a BMI of <16 kg/m$^2$ for the definition of severe malnutrition. This decision was taken since a considerable percentage of patients was prostrate at admission and unable to stand on a scale.

**Data sources.** Data for this study was acquired from IPD and OPD client level databases containing information obtained during routine care. Patient information was entered by clinicians into in- and out-patient files and encoded into dedicated data bases. A custom-made Excel-based database was used for patients hospitalised at Donka and a Tier.net® database was used for patient follow up in OPD. Each patient has a unique ID code used in both the OPD and IPD databases that allowed patient data linkage. After merging the databases, the pre-anonymised data was cleaned, and any key missing data were checked with the patient files by the in-project epidemiologist (REH) or the attending physician (SLA), and data were entered into the study database if it could be ascertained from the patient files. All databases were de-identified post data cleaning and after merging by removing patient codes and generating a new study code (studyid). All data was stored in password protected files, and from this point forward de-identified information only was accessed by the study investigators. Patients' medical records were accessed between June 2016 until December 2017.

**Statistical analysis.** Baseline socio-demographic, clinical characteristics and outcomes for hospitalized patients, and patients who could be followed up post-discharge were described using frequencies and proportions for categorical data and medians with corresponding interquartile ranges (IQR) for skewed continuous variables. Outcomes of hospitalization were classified as alive or dead at discharge. Morbidity data was classified as reported above. Furthermore, we calculated frequencies of ART initiation and switch among naïve and experienced patients, respectively as well as the proportion of ART experienced patients who fulfilled criteria for switch based on either one or two viral loads. Patients were followed after hospitalization if they were discharged alive, and their hospital ID could be linked to the OPD database.

Long-term outcomes were presented from date of discharge from hospital as the start date. Any patient who was lost at the day of exit was assigned one day of follow up time. Outcomes during follow up were classified as alive (including patients referred to other facilities at study censor), dead or LTFU which was defined as missing the last scheduled clinic appointment by a period of 90 days, or 90 days since hospitalisation if the patient was lost immediately after hospitalisation. Tiernet would automatically classify patients as LTFU if their appointment was missed, and project staff would actively follow up patients who were classified as LTFU to encourage them to come back to care, or to confirm their outcome classification. We also included the outcome attrition, which combined LTFU and dead. Chi squared tests and fisher's exact tests were used as appropriate to test for differences between alive, dead and LTFU by categorical baseline characteristics. Chi squared tests were carried out with and without missing data in order to understand if the missing data might affect the analyses. Comparisons of medians by the three outcomes were calculated using k-sample comparison of medians. A two-sided alpha <0.025 threshold was considered as statistically significant for all analyses. All analyses were conducted with STATA version 15, and data cleaning was conducted in Microsoft Excel 2016.

**Ethical committee approval.** The study was approved by the national ethics committee for health research (CNERS) reference 113/CNERS/2018 in Guinea who waived the requirement for informed consent. Also, the study fulfilled the exemption criteria set by the doctors without borders Ethics Review Board for a posteriori analyses of routinely collected clinical data and thus did not require MSF ERB review. It was conducted with permission from the Medical Director, Operational Centre Brussels, Belgium. As an a posteriori analyses of routinely collected clinical data, no individual consent was sought for this study.

## Results

A total of 401 HIV patients were hospitalized between August 2017 and April 2018. Of them, 194 were followed-up for 6 months after hospitalization. Fig 1 shows discharge and follow up outcomes.

### Baseline demographic and clinical characteristics

Tables 1 and 2 show demographics and clinical characteristics at admission of the hospitalised cohort, and the cohort, that was followed up after hospital discharge. Baseline characteristics of the hospitalised cohort and the follow-up cohort were similar. There was no significant difference between the two groups in regards to age (36 IQR:28–45 vs. 35 IQR:25–43, p = 0.684), sex (230 (57%) females vs. 112 (58%) female, p = 0.931), CD4 count (64 IQR:24–187 vs. 80 IQR:28–208, p = 0.128) or distribution of VL (171 (43%) vs. 96 (49%) with VL>1000, p = 0.943). 161 (40%) patients reported no prior treatment, while 125 (31%) reported that they had been on ART for more than 6 months. More patients in the hospitalised cohort were ART naïve compared to those in the follow up cohort (161 (40%) vs. 51 (26%), p<0.05). Among both cohorts, the number of patients with exposure to 1$^{st}$ line ART for less than 6 months was similar 125 (31%) vs. 54 (28%) p = 0.54, respectively. At admission therefore, the two cohorts had comparable ART characteristics.

### Patient outcomes

Table 1 shows the overall outcomes of patients during hospitalisation. 143 (36%) patients died during hospitalisation, and of these, 18 (13%) died within the first 48 hours of admission. Median time until death at 1$^{st}$ hospitalization was 6 days (IQR:3–13). Median length of hospitalization for patients who survived was 13 (IQR:7–20) days. Of all major opportunistic

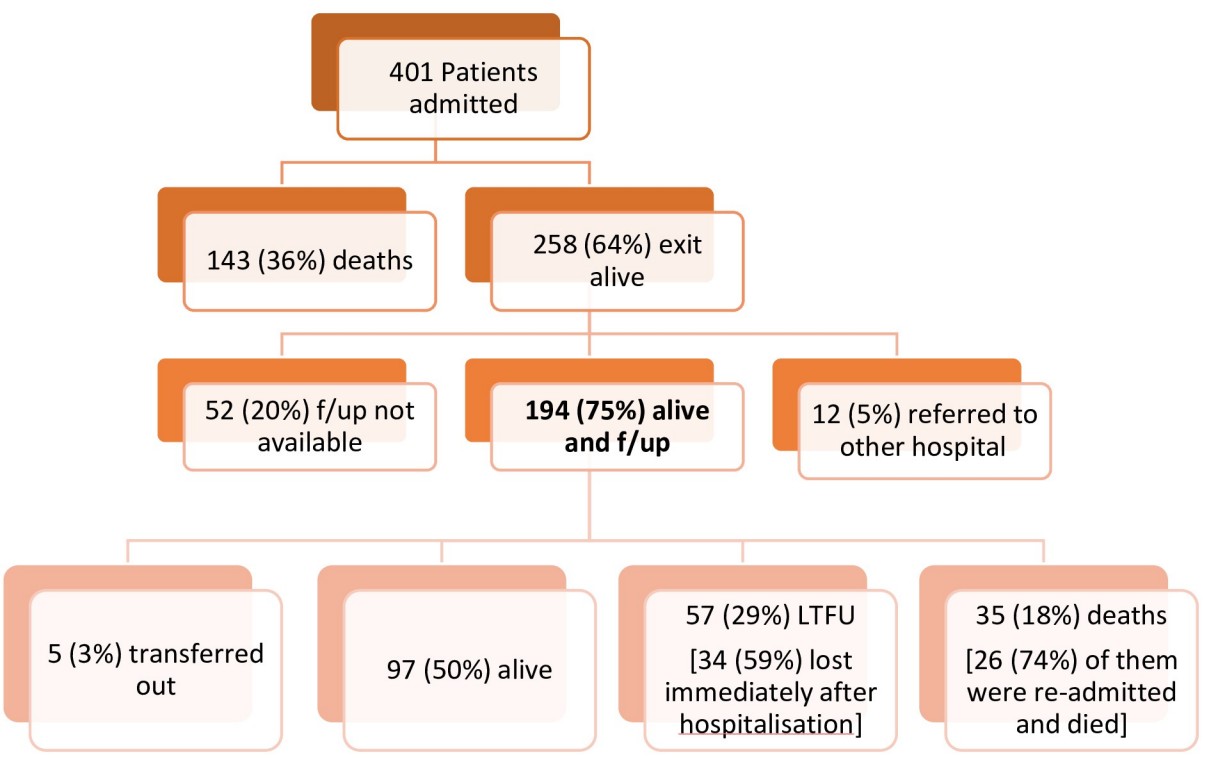

**Fig 1. Outcome of 401 patients admitted to USFR, Donka, Conakry, Guinea between 1ˢᵗ August 2017 and 31ˢᵗ April 2018.**

infections, patients with Cryptococcal meningitis had the longest hospitalizations with a median length of 21 days (IQR:20–30).

The outcomes of all patients at the end of the 6 month follow up period, are shown in Table 2: 102 (53%) remained active, with 5 among them being censored as they were transferred out of the cohort, 57 (29%) were LTFU and 35 (18%) died. Of those that were lost to follow up, 34 (59%) were lost immediately after an IPD episode, and never returned for their follow-up OPD appointment which were routinely scheduled 3–5 days after hospitalisation. Overall, 67 (35%) were re-hospitalized once or twice and 22 (11%) were re-hospitalized at least 3 times. Of those re-hospitalized 26 (47%) died in-hospital. The median time to rehospitalization at USFR was 55 days (IQR:32–99).

Patients who died during hospitalization (48 cells/mm$^3$ (IQR:15–112) vs 75 cells/mm$^3$ (IQR:28–208) p = 0.018) or during follow up (33 cells/mm$^3$ (IQR:13–98 vs 80 cells/mm$^3$ (IQR:32–202) p = 0.038) had lower CD4 counts than those that survived. Patients who were LTFU however had a higher median CD4 count than those who survived (131cells/mm$^3$ (IQR:29–278)).

Patients with a documented treatment interruption had comparable outcomes during hospitalization to patients without a documented treatment interruption (33 (34%) vs 81 (39%) p = 0.425) mortality, respectively. However, during follow up those with a treatment interruption had higher level of attrition (27 (63%) vs 45 (45%) p = 0.033) than patients with no treatment interruption. Patients on second line (44%) or who had been on first line for more than 6 months (57%) were more likely to have a recorded treatment interruption than those who had been on treatment for less than 6 months (37%) (p<0.001). Outcomes for patients at hospital discharge and during follow up were similar for ARV naïve and experienced patients (p = 0.726, p = 0.440).

**Table 1. Clinical and demographic characteristics and in-hospital outcomes of the 401 patients hospitalised between August 2017 and April 2018.**

| Variable | | Total | | Alive | | Died | | P value Without missing / With missing for chi2** |
|---|---|---|---|---|---|---|---|---|
| | | N | % (col) | N | % (row) | N | % (row) | |
| Total patients included | | 401 | 100 | 258 | 74% | 143 | 36 | |
| Age | Median (IQR) | 36 (28–45) | | 35 (25–43) | | 40 (30–47) | | 0.009 |
| Sex | Female | 230 | 57 | 149 | 64% | 81 | 35 | 0.830 |
| | Male | 171 | 43 | 109 | 63% | 62 | 36 | |
| CD4 in cells/mm$^3$ at admission | Median (IQR) | 64 (24–187) | | 75 (28–208) | | 48 (15–112) | | 0.018 |
| Virological status * | TND | 42 | 10 | 29 | 69% | 13 | 31 | 0.100 / 0.006 |
| | 1+ VL 40–999 | 41 | 10 | 22 | 54% | 19 | 46 | |
| | 1VL> = 1000 | 166 | 41 | 122 | 73% | 44 | 27 | |
| | 2VL >1000 | 16 | 4 | 10 | 63% | 6 | 38 | |
| | 2VL 1st >1000 2nd <1000 | 3 | 1 | 3 | 100% | 0 | 0 | |
| | No VL result | 133 | 33 | 72 | 54% | 61 | 46 | |
| ARV status at admission | Not initiated | 161 | 40 | 108 | 67% | 53 | 33 | 0.726 / 0.019 |
| | 1st line <6 months | 86 | 21 | 52 | 60% | 34 | 40 | |
| | 1st line >6 months | 125 | 31 | 84 | 67% | 41 | 33 | |
| | 2nd line | 18 | 4 | 12 | 67% | 6 | 33 | |
| | Missing | 11 | 3 | 2 | 18% | 9 | 82 | |
| Treatment interruption | Yes | 97 | 24 | 64 | 66% | 33 | 34 | 0.425 / 0.354 |
| | No | 209 | 52 | 128 | 61% | 81 | 39 | |
| | Missing | 95 | 24 | 66 | 61% | 29 | 31 | |
| At least one neurological sign | Yes | 127 | 32 | 71 | 56% | 56 | 44 | 0.092 / 0.012 |
| | No | 270 | 67 | 186 | 68% | 84 | 31 | |
| | Missing | 4 | 1 | 1 | 68% | 3 | 75 | |
| Symptoms for at least 2 weeks before hospitalisation | No | 90 | 22 | 67 | 74% | 23 | 26 | 0.033 / 0.062 |
| | Yes | 269 | 67 | 167 | 62% | 102 | 38 | |
| | Missing | 42 | 10 | 24 | 62% | 18 | 43 | |

*This variable includes all VL results available for a patient which were either before or at admission for the 401 patients analysed during hospitalisation

** For Chi squared tests, results including and excluding missing data have been included, with the first result shown that without the missing data. Only one result is presented if there is no missing data for that variable (sex). The Chi squared test compares patients who are alive and dead. Patients who remained alive are not presented, in order to save space. A total column is presented instead in order to show the prevalence of each factor in the cohort.

Mortality rates during hospitalization and attrition post-hospitalization stratified by viral load status were similar. Patients who had had only undetectable viral loads recorded either during, before or after hospitalization, had better outcomes overall. As shown in Tables 1 and 2, patients who had no VL recorded or who had any number of VL recorded above undetectable (taken to be above 40 copies/ml) had poor outcomes, regardless of ARV regimen or switch to second line.

## ARV initiation and treatment failure

At hospital admission, 161 (40%) patients reported to be treatment naïve, 56 (35%) of which were initiated on 1st line ART during hospitalization.

171 (43%) of the 401 hospitalized patients and 96 (49%) of the 194 patients followed up post-discharge had one viral load above 1000 copies/ml recorded at admission, 16 (4%) and 30 (15%) patients had two VL>1000 recorded during or before hospitalization, respectively. On the basis of two elevated viral loads among patients with 1st line ART exposure>6 months, only 1 (9%) out of 11 patients eligible for switch to 2nd line ART was switched during

**Table 2. Clinical and demographic characteristics and post-discharge outcomes of the 194 patients who could be followed up after hospitalisation between August 2017 and April 2018.**

| Variable | | Total | | LTFU | | Dead | | Attrition (dead +LTFU) | | Active | | P value* |
|---|---|---|---|---|---|---|---|---|---|---|---|---|
| | | N | % (col) | N | % (row) | N | % (row) | N | % (row) | N | % (row) | |
| Total patients included | | 194 | 100 | 57 | 29 | 35 | 18 | 92 | 47 | 97 | 50% | |
| Age | Median (IQR) | 35 (25–43) | | 36 (28–48) | | 38 (33–50) | | 37 (30–48) | | 30 (23–40) | | 0.030 |
| Sex | Female | 112 | 58 | 29 | 26 | 20 | 18 | 49 | 44 | 61 | 54% | 0.423 |
| | Male | 82 | 42 | 28 | 34 | 15 | 18 | 43 | 52 | 36 | 44% | |
| CD4 in cells/mm$^3$ at admission | Median (IQR) | 80 (28–208) | | 131 (29–278) | | 33 (13–98) | | 75 (22–210) | | 80 (32–202) | | 0.017 |
| Virological status ** | VL TND | 26 | 13 | 6 | 23 | 1 | 4 | 7 | 27 | 19 | 73% | 0.011 / 0.001 |
| | 1+ VL 40–999 | 25 | 13 | 8 | 32 | 8 | 32 | 16 | 64 | 8 | 32% | |
| | 1VL> = 1000 | 69 | 36 | 27 | 39 | 10 | 15 | 37 | 54 | 31 | 45% | |
| | 2VL >1000 | 30 | 15 | 3 | 10 | 8 | 27 | 11 | 37 | 19 | 63% | |
| | 2VL 1st >1000 2nd <1000 | 19 | 10 | 2 | 11 | 6 | 32 | 8 | 42 | 11 | 58% | |
| | No VL result | 25 | 13 | 11 | 19 | 2 | 6 | 13 | 25 | 9 | 36% | |
| ARV status at hospital exit | Not initiated | 51 | 26 | 12 | 24 | 10 | 20 | 32 | 44 | 29 | 57% | 0.657 |
| | 1st line <6 months | 74 | 38 | 23 | 31 | 14 | 19 | 37 | 50 | 37 | 50% | |
| | 1st line >6 months | 54 | 28 | 19 | 36 | 6 | 11 | 25 | 47 | 29 | 54% | |
| | 2nd line | 15 | 8 | 3 | 20 | 5 | 33 | 8 | 53 | 7 | 47% | |
| | Missing | 0 | 0 | 0 | 0 | 0 | 0 | 0 | 0 | 0 | 0% | |
| Treatment interruption | Yes | 43 | 22 | 15 | 35 | 12 | 28 | 27 | 63 | 15 | 35% | 0.208 / 0.033 |
| | No | 100 | 52 | 25 | 25 | 20 | 20 | 45 | 45 | 54 | 54% | |
| | Missing | 51 | 26 | 17 | 33 | 3 | 6 | 20 | 39 | 28 | 55% | |
| At least one neurological sign | Yes | 45 | 23 | 15 | 33 | 8 | 18 | 23 | 51 | 22 | 49% | 0.608 / 0.379 |
| | No | 148 | 76 | 42 | 28 | 26 | 18 | 68 | 46 | 75 | 51% | |
| | Missing | 1 | 1 | 0 | 0 | 1 | 100 | 1 | 100 | 0 | 0% | |
| Symptoms for at least 2 weeks before hospitalisation | No | 50 | 26 | 15 | 30 | 7 | 14 | 22 | 44 | 26 | 52% | 0.626 / 0.904 |
| | Yes | 125 | 64 | 37 | 30 | 24 | 19 | 61 | 49 | 62 | 50% | |
| | Missing | 19 | 10 | 5 | 26 | 4 | 21 | 9 | 47 | 9 | 47% | |

* The p value refers to chi squared tests comparing all 3 outcomes (active, LTFU, dead) but patients who remained alive are not presented, in order to save space. A total column is presented instead in order to show the prevalence of each factor in the cohort. For CD4 and age the p value is from a K-sample equality of medians test. For Chi squared tests, results including and excluding missing data have been included, with the first result shown that without the missing data. Only one result is presented if there is no missing data for that variable (sex)

**This variable includes all VL results available done either before, or at admission or during follow up

hospitalization, and one more patient was switched during follow up. Using only one increased viral load as definition of failure among ART experienced patients, 37 patients would have been eligible for immediate switch during their hospitalization. Of those, only 4 (11%) were switched during hospitalization and were all discharged alive. Out of the remaining 33 patients who were not switched, 8 (25%) died during hospitalization with a median time until death of 4.5 days (IQR:2.5–8). During the follow up period, a further 6 out of 25 (24%) were switched. At the end of the study period, 3 out of 6 (50%) among those that were switched and 15 out of 23 (70%) among those that were not, were no longer in care.

Of the 11 patients on second line treatment, none were switched to third line, although 3 (27%) did have two VL recorded above 1000 copies/ml.

## Morbidity and mortality

Tables 3 and 4 show major opportunistic infections (OIs) and associated conditions at exit for hospitalized patients and those followed up post-discharge. 84 (21%) patients presented with a haemoglobin level of below 7g/dl and 137 (34%) patients presented with severe malnutrition. Tuberculosis (TB) was the most frequent clinical diagnosis at exit with 295 (74%) diagnoses as well as the leading cause of mortality with 101 deaths (70%). 123 (31%) patients showed signs of disseminated TB, 63 (16%) were diagnosed with TB meningitis. 52 (13%) and 36 (9%) TB cases were diagnosed without biological confirmation and already on treatment at admission, respectively. Of the 295 patients with a primary diagnosis of TB, 68 (23%) had no other diagnosis. Additional diagnosis amongst patients with TB were: Toxoplasmosis (30%), diarrhoea (19%), oesophageal candidiasis (19%), PCP (16%) and Cryptococcal meningitis (CM) (6%). Out of 55 patients that were hospitalized at USFR for a 2nd time, 20 (36%) had a different diagnosis. One case who initially was recorded as a case of chronic diarrhoea was diagnosed with CM upon 2nd hospitalization, while two TB cases were diagnosed with Toxoplasmosis during their 2nd hospitalization. Tables 3 and 4 show that rates of death and attrition during and after hospitalization are similar for all OIs. One exception is that patients with signs of severe acute kidney injury had worse outcomes than patients with other conditions with 23 (57%) $p = 0.002$ dying during hospitalization, and 7 (50%) $p = 0.650$ being LTFU or dead during the 6 months of follow up.

**Table 3. Opportunistic infections and morbidity characteristics of the 401 patients hospitalised between August 2017 and April 2018.**

| Variable | | Total N = 401 | | Alive N = 258 | | Died N = 143 | | P value** |
|---|---|---|---|---|---|---|---|---|
| | | N | % (col) | N | % (col) | N | % (row) | |
| Major OIs/ causes of death | Pulmonary TB | 24 | 6 | 18 | 75% | 6 | 25 | 0.261 |
| | Disseminated TB | 123 | 31 | 82 | 67% | 41 | 33 | 0.517 |
| | Clinical TB | 52 | 13 | 36 | 69% | 16 | 31 | 0.430 |
| | TB Meningitis | 63 | 16 | 39 | 62% | 24 | 38 | 0.660 |
| | Drug resistant TB | 8 | 2 | 7 | 88% | 1 | 12 | 0.141 |
| | TB on treatment | 36 | 9 | 23 | 64% | 13 | 37 | 0.785 |
| | Toxoplasmosis | 55 | 14 | 38 | 70% | 17 | 31 | 0.428 |
| | Crypto disease (CRAG serum+ and Crag CSF-or missing) | 16 | 4 | 12 | 75% | 4 | 25 | 0.337 |
| | Crypto meningitis (Crag CSF+) | 17 | 4 | 9 | 53% | 8 | 47 | 0.625 |
| | PCP | 14 | 3 | 6 | 43% | 8 | 57 | 0.088 |
| | Chronic diarrhoea | 30 | 7 | 20 | 66% | 10 | 33 | 0.782 |
| | Karposi Sarcoma | 9 | 2 | 7 | 77% | 2 | 22 | 0.395 |
| | Oesophageal Candida | 34 | 8 | 32 | 94% | 2 | 6 | <0.001 |
| | Community acquired Pneumonia | 8 | 2 | 4 | 50% | 4 | 50 | 0.392 |
| | Malaria | 11 | 3 | 10 | 91% | 1 | 9 | 0.062 |
| Major associated conditions (N = 365) | Community acquired blood stream infection | 34 | 9 | 6 | 18% | 28 | 82 | <0.001 |
| | Anaemia<7g/dl | 84 | 21 | 53 | 63% | 31 | 37 | 0.682 |
| | Renal impairment* | 40 | 10 | 17 | 43% | 23 | 58 | 0.002 |
| | Malnutrition (MUAC <210mm) | 137 | 34 | 80 | 58% | 57 | 42 | 0.122 |
| | Liver impairment | 14 | 4 | 7 | 50% | 7 | 50 | 0.254 |

* Defined as severe acute kidney failure, unresponsive to correction of hypovolemia

** The chi squared test compares dead with alive patients. Patients who remained alive are not presented, in order to save space. A total column is presented instead in order to show the prevalence of each factor in the cohort.

**Table 4. Opportunistic infections and morbidity characteristics of the 194 patients who could be followed up after hospitalisation between August 2017 and April 2018.**

| Variable | | Total | | LTFU | | Dead | | Attrition (dead +LTFU) | | Alive and in care | | P value* |
|---|---|---|---|---|---|---|---|---|---|---|---|---|
| | | N | % (col) | N | % (row) | N | % (row) | N | % (row) | N | % (row) | |
| Total patients included | | 194 | 100 | 57 | 29 | 35 | 18 | 92 | 47 | 97 | 50% | |
| Major OIs at hospital exit | Pulmonary TB | 9 | 5 | 3 | 33 | 1 | 11 | 4 | 44 | 6 | 60% | 0.013 |
| | Disseminated TB | 57 | 29 | 16 | 28 | 18 | 32 | 34 | 60 | 22 | 39% | |
| | Clinical TB | 26 | 13 | 7 | 27 | 2 | 8 | 9 | 35 | 14 | 54% | |
| | TB Meningitis | 25 | 13 | 8 | 32 | 8 | 32 | 17 | 64 | 9 | 35% | |
| | Drug resistant TB | 0 | 0 | 0 | 0 | 0 | 0 | 0 | 0 | 0 | | |
| | TB on treatment | 17 | 9 | 6 | 35 | 2 | 12 | 8 | 47 | 10 | 56% | |
| | Toxoplasmosis | 25 | 13 | 9 | 36 | 5 | 20 | 14 | 56 | 11 | 44% | 0.697 |
| | Crypto disease (CRAG serum+ and Crag CSF-or missing) | 10 | 5 | 3 | 30 | 2 | 20 | 5 | 50 | 5 | 50% | 0.700 |
| | Crypto meningitis (Crag CSF+) | 8 | 4 | 0 | 0 | 1 | 13 | 1 | 13 | 7 | 88% | |
| | PCP | 13 | 7 | 3 | 23 | 1 | 8 | 4 | 31 | 3 | 60% | 0.493 |
| | Chronic diarrhoea | 18 | 9 | 5 | 28 | 2 | 11 | 7 | 39 | 11 | 61% | 0.672 |
| | Karposi Sarcoma | 7 | 4 | 6 | 86 | 0 | 0 | 6 | 86 | 1 | 14% | 0.011 |
| | Oesophageal Candida | 27 | 14 | 9 | 33 | 6 | 22 | 15 | 55 | 12 | 44% | 0.689 |
| | Community acquired Pneumonia | 4 | 2 | 0 | 0 | 0 | 0 | 0 | 0 | 3 | 75% | 0.015 |
| | Malaria | 5 | 3 | 1 | 20 | 0 | 0 | 1 | 20 | 4 | 80% | 0.545 |
| Major associated conditions | Community acquired blood stream infection | 4 | 2 | 1 | 25 | 1 | 25 | 2 | 50 | 2 | 50% | 0.971 |
| | Anaemia<7g/dl | 41 | 21 | 10 | 24 | 9 | 22 | 19 | 46 | 21 | 51% | 0.825 |
| | Renal impairment** | 14 | 7 | 3 | 21 | 4 | 29 | 7 | 50 | 7 | 50% | 0.650 |
| | Malnutrition (MUAC<210mm) | 64 | 33 | 16 | 25 | 13 | 20 | 29 | 45 | 82 | 49% | 0.467 |
| | Liver impairment | 5 | 3 | 2 | 40 | 0 | 0 | 2 | 40 | 3 | 60% | 0.715 |

* The p value refers to chi squared tests comparing all 3 outcomes (active, LTFU, dead). Patients who remained alive are not presented, in order to save space. A total column is presented instead in order to show the prevalence of each factor in the cohort.

** Defined as severe acute kidney failure, unresponsive to correction of hypovolemia

## Discussion

To our knowledge, this is the first study reporting post-hospitalisation survival data in HIV patients in a West African context. Survival rates in our cohort were extremely low.

Whilst mortality rates for patients with advanced HIV disease are generally poor across the published literature, outcomes for this cohort are amongst the worst and are not substantially different to outcomes of the pre-cART era globally [15]. Contemporary studies from Kinshasa [16], Ethiopia [17], Ghana [18] and Senegal [19] report similarly poor in-hospital mortality of HIV positive patients with 30%, 29%, 40% and 44% respectively. A study published in 2019 in Zambia showed better outcomes with 5.4% and 22% mortality at discharge and 3 months post-discharge respectively, though this study excluded critically ill patients [20].

In our cohort, 40% and 35% of patients presenting with advanced disease were ART naïve and had been on ART for more than six months, respectively. Awareness of serostatus among HIV positive men and overall ART coverage in Guinea are at 37% and 35%, respectively [5, 21] indicating that the global shift in the HIV epidemic from ART naïve towards 1st line experienced patients has not happened to the same extent in Guinea [10]. Although the true number of treatment experienced patients might be underestimated due to patients underreporting

previous exposure to ARV due to shame and stigma, this finding has important implications. For one, continued efforts to improve HIV screening and ART coverage can go a long way to prevent advanced HIV in Guinea. Second, NNRTI based 1[st] line regimens seem to continue to be a viable option, especially given the low rates of primary resistance in this setting [12].

## Screening and management of co-morbidities

More than two thirds (73%) of patients were treated for TB. 67% of patients had symptoms for at least 2 weeks, indicating late presentation and or missed chances for earlier detection at OPD level. Almost 80% of patients with a TB diagnosis presented additional co-morbidities. Most additional diagnoses were made based on clinical suspicion which might have led to over-treatment. While potentially resulting in treatment toxicity, empirical treatment is advisable in a context of imperfect diagnostic tools and conditions associated with high mortality. Provision of comprehensive screening and treatment packages in compliance with recent WHO recommendations should be further encouraged [3, 7].

Of note, mortality and attrition were particularly high for patients with severe kidney injury. In our cohort however, end-organ failure must be regarded as a proxy of underlying sepsis rather than a standalone condition.

## Screening for treatment failure

In this study we applied two different definitions for ART failure and eligibility for switch from 1[st] to 2[nd] line of ART. Following the definition that was in place during most of the study time period, requiring 2 VL above 1000, only 11 patients would have been eligible. Only two (18%) of those eligible under this definition were switched. This means that out of 125 critically ill patients who were clinically failing while on 1[st] line ART for more than 6 months only 9% would have qualified for switch to 2[nd] line therapy and as few as 2% were changed. Using an approach where severely sick patients with one raised VL would be considered as ART failures in need of a switch of ART regimen, only 37 out of 125 (30%) patients would have met eligibility criteria for a switch to 2[nd] line ART.

It is well known that delayed switch is common and increases mortality among patients with advanced disease [22]. Reasons for delayed switch include concern for side effects and pill burden, complicated protocols as well as organisational challenges such as poor access to laboratory monitoring [23]. Our data suggests that at this moment, switch from 1[st] to 2[nd] line ARVs in severely ill patients in low resource settings is excessively difficult and rarely done in practice. The study was conducted in a setting with relatively good access to laboratory monitoring so this alone was not considered a principle hindrance, but that more complex barriers to implementation existed that could be elucidated with further research. With increasing rates of ART experienced patients among critically ill HIV-patients and high rates of acquired drug resistance among them [8, 16, 24], delayed switch will continue to be an important contributor to mortality. Recent changes in WHO guidelines for ART switch in critically ill HIV-positive patients based on just one VL or on clinical grounds alone are an important first step to address this issue [7, 8].

## Improvements in psychosocial care

In our study, attrition was similar for all patient groups, regardless of switch to second line or of the type and treatment of OIs. Furthermore, 59% of patients who were lost to follow up disappeared immediately post-discharge without any further contact with OPD, nullifying any potential effect that prompt and urgent care as well as timely change to 2[nd] line ART might have had. These results indicate that the prompt and effective treatment provided in a

specialised car unit was not sufficient for improving treatment outcomes, and there may be other factors at play. The difficulties experienced by hospitalised in-patients in West and Central Africa have been described in a qualitative study from Kinshasa which showed that patients experienced serious mental and social issues [25]. Patients often felt highly stigmatised and hopeless about their situation and many felt unable to disclose their status or ask for help and support from their families. Furthermore, patients with HIV in Guinea Conakry do not always know where their next meal will be coming from after discharged. We therefore urge caution in transferring the results from the study by Shroufi et al. to our setting without further consideration. In this study, the authors estimate that switching from 1st to 2nd line treatment on the basis of 1 VL would in itself result in more than 10.000 prevented deaths per year in South Africa [24]. We agree that rapid switch to 2nd line is paramount for the survival of patients who fail 1st line ART. However, we hypothesize that measures such as suggested by Shroufi et al. can only draw on their full potential when combined with intensified patient support to facilitate transition to ambulatory care. In their 2017 guideline, WHO recommends psychosocial support after ART initiation in patients with advanced HIV, in order to support patients deal with the difficulties they experience [7]. Evidence of the impact of comprehensive, tailored psychosocial and economic programmes such as support groups, individual or group counselling, cash transfers, and interventions that reduce transport difficulties such as integration, task-shifting or decentralisation have been well described [26–32] For example data from 2015 reported a 28% reduction in mortality when a medical intervention was combined with home visits in advanced HIV patients starting ART [33].

## Limitations

The major limitation to this study is that as a retrospective analysis of routine data, there were challenges with data quality. 64 (25%) of patients who left hospital alive were not followed up–either due to being referred to another hospital (12–5%) or due to data entry errors in their ID code (9–3%), or due to being from a different cohort before admission, with a different ID numbering system 43 (17%). Multivariate analyses of both hospital and follow-up outcomes did not reveal useful information beyond known associations of advanced disease with mortality and were therefore not included. Information on clinical diagnosis used for this study underestimates the true number of diagnoses per patient as the database only recorded two co-morbidities. Other non-HIV related diseases may be under notified as OIs were normally prioritized as diagnosis. For some variables (e.g. HIV-VL, CD4), the proportion of missing data was high, especially in patients who died rapidly after admission. Finally, the actual outcome of those LTFU remains uncertain due to the lack of full follow up, as the follow up system only allowed active follow up following a missed, scheduled outpatient appointment, and not always following discharge from hospital as the outpatient monitoring system was separated to the in-patient system. Improvements to the follow up system were made after this study was conducted. Prior research on the issue reports mortality rates among patients LTFU between 12% and 87% [24, 33–35]. Given the precarious health of patients at discharge as well as high barriers to treatment in our setting, we must assume a high mortality rate among patients LTFU.

## Conclusions

Outcomes for critically ill HIV-positive patients hospitalized in Guinea, Conakry were poor during hospitalization and follow-up. We estimate 1-in-3 patients to be alive and in care 6 months post-discharge. This study highlights the severe nature of advanced HIV in a resource limited, low prevalence context and identifies attrition during transition from hospital to outpatient services as an important pitfall in patients care continuum.

## Supporting information

**S1 Database.**
(XLSX)

## Acknowledgments

We would like to thank the patients and their families hospitalized at USFR, the staff at USFR and the Ministry of Health for supporting this study.

## Author Contributions

**Conceptualization:** Sebastian Ludwig Albus, Rebecca E. Harrison, Esther C. Casas.

**Data curation:** Rebecca E. Harrison, Ramzia Moudachirou, Esther C. Casas, Abdourahimi Diallo.

**Formal analysis:** Sebastian Ludwig Albus, Rebecca E. Harrison, Ramzia Moudachirou, Abdourahimi Diallo.

**Investigation:** Sebastian Ludwig Albus, Rebecca E. Harrison, Issiaga Camara, Marie Doumbuya.

**Methodology:** Rebecca E. Harrison, Esther C. Casas.

**Project administration:** Kassi Nanan-N'Zeth.

**Software:** Abdourahimi Diallo.

**Supervision:** Kassi Nanan-N'Zeth, Benoit Haba, Esther C. Casas, Petros Isaakidis, Fode Bangaly Sako, Mohammed Cisse.

**Writing – original draft:** Sebastian Ludwig Albus, Rebecca E. Harrison.

**Writing – review & editing:** Sebastian Ludwig Albus, Rebecca E. Harrison, Ramzia Moudachirou, Kassi Nanan-N'Zeth, Benoit Haba, Esther C. Casas, Petros Isaakidis, Issiaga Camara, Marie Doumbuya, Fode Bangaly Sako.

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
