## [Decision Letter · Decision Letter 0]

11 Mar 2021

PONE-D-20-31177

Poor outcomes among critically ill HIV-positive patients at hospital discharge and post-discharge in Guinea, Conakry: A retrospective cohort study

PLOS ONE

Dear Dr. Albus,

Thank you for submitting your manuscript to PLOS ONE. After careful consideration, we feel that it has merit but does not fully meet PLOS ONE’s publication criteria as it currently stands. Therefore, we invite you to submit a revised version of the manuscript that addresses the points raised during the review process.

The manuscript has been evaluated by two reviewers, and their comments are available below.

The reviewers have raised a number of concerns that need attention. They request additional information on the rationale and methodological aspects of the study, as well as revisions to the discussion of the results. They also request improvements in the presentation and copyediting of the manuscript.

Could you please revise the manuscript to carefully address the concerns raised?

We look forward to receiving your revised manuscript.

Kind regards,

Marianne Clemence

Associate Editor

PLOS ONE

Journal Requirements:

2. You indicated that ethical approval was not necessary for your study. We understand that the framework for ethical oversight requirements for studies of this type may differ depending on the setting and we would appreciate some further clarification regarding your research. Could you please provide confirmation from your institutional review board or research ethics committee (e.g., in the form of a letter or email correspondence) that ethics review was not necessary for this study? Please include a copy of the correspondence as an "Other" file.

Additionally, in the ethics statement in the manuscript and in the online submission form, please provide additional information about the patient records used in your retrospective study, including:

a) whether all data were fully anonymized before you accessed them;

b) the date range (month and year) during which patients' medical records were accessed.

3. To comply with PLOS ONE submission guidelines, in your Methods section, please provide additional information regarding your statistical analyses.

For more information on PLOS ONE's expectations for statistical reporting, please see https://journals.plos.org/plosone/s/submission-guidelines.#loc-statistical-reporting

Additional Editor Comments:

We note that one or more reviewers has recommended that you cite specific previously published works. As always, we recommend that you please review and evaluate the requested works to determine whether they are relevant and should be cited. It is not a requirement to cite these works. We appreciate your attention to this request.

Reviewers' comments:

Reviewer's Responses to Questions

**Comments to the Author**

1. Is the manuscript technically sound, and do the data support the conclusions?

Reviewer #1: Partly

Reviewer #2: Yes

2. Has the statistical analysis been performed appropriately and rigorously? 

Reviewer #1: No

Reviewer #2: Yes

3. Have the authors made all data underlying the findings in their manuscript fully available?

Reviewer #1: No

Reviewer #2: Yes

4. Is the manuscript presented in an intelligible fashion and written in standard English?

Reviewer #1: No

Reviewer #2: No

5. Review Comments to the Author

Reviewer #1: First of all I would like to thank for the opportunity to review this manuscript. The authors determined the characteristics and outcomes of critically ill HIV-positive patients admitted to the hospital. However, the study suffers from a number of weaknesses. I recommend the authors to include the following comments in their manuscript to make it more suitable for publication.

Introduction

1. The authors should clearly describe the statement of problem and significance of the study in last paragraph of the introductory part.

2. Write abbreviations in full when it first appears (E.g. MSF).

Methods

1. Under this section, the authors need to include a number of sub topics which includes, sample size determination and sampling technique, data collection procedure and quality assurance, patient follow-up and outcome measurement and outcome validation

2. Line 57-“Data for this study was…” Data is plural. So, use were instead of was.

3. Line 63- Could the authors explain how they managed missing data?

4. Line 70-71- I suggest the Outcome definitions to put it under “outcome validation” separately.

5. The authors should provide the reference number for Ethical Committee Approval.

Results

1. Table: 1 is not easy to reader. Consider revising it. Revise also the topic,” Clinical characteristics and in-hospital outcomes..” since the table includes socio-demographic data.

2. Line 160-consider revising the sentence.

3. The authors should include the combined antiretroviral therapy (cART) regimens of the patients in a Table.

4. It was of interest if the authors could include the risk factors (predictors) of in-hospital mortality. This would help in identifying and tackling those risk factors. It is also mentioned under limitation as well.

5. Consider defining terminologies like malnutrition, and etc.

6. Could the authors determine the leading cause of death in the cohort?

Discussion

Your discussion is nice. Could you consider comparing the in-hospital death of AIDS patients in your study with those conducted in Eastern Africa (E.g. Negera GZ, https://doi.org/10.1371/journal.pone.0226683).

Reviewer #2: “Poor outcomes among critically ill HIV-positive patients at hospital discharge and post discharge in Guinea, Conakry: A retrospective cohort study” is a very informative and interesting paper that needs to fill some gaps before being published, in my opinion

The main gap is in the discussion of results, when the authors realized that the key point is the model of care which is clearly inadequate to the real world in Guinea. For example the high LTFU could be due simply to the lack of money for transport: however this information is not reported by the study as well as others and this should be reported as a limitation (which has been done only partially when the authors state “.. Finally, the actual outcome of those LTFU remains 271 uncertain due to the lack of active follow up.. “).

There is large number of papers who discuss the relation between social support joined with access to the standard of HAART care. I encourage the author to go in deep with this issue that that seems to be the key to improve HIV care in LRS

Other major gaps are related to the main one: the relevance of doling viral load on a routine basis to address the gap in timely switching to second line for example: is this a matter of the model of care or just the lack of reagents and laboratory able to perform it? To my knowledge of guinea (and especially Conakry) the laboratory should have enough potential capacity of performing Viral loads. However this should be discussed

Minor observations

Tables are some time so rich to became difficult to be read. The authors should consider the possibility of shifting part of data in a supplementary section in order to facilitate the readers in getting the key information

Please pay attention to percentage that in several cases do not round up to 100 (table 1 column “total” Row “VL cat copies/ml…..) for example

Some statements in the text are too much detailed, maybe

“..One case who initially was recorded as a case of chronic diarrhoea was diagnosed

with CM upon 2nd 175 hospitalization, while two TB cases were diagnosed with Toxoplasmosis

during their 2nd 176 hospitalization. Table 3 and 4 show that rates of death and attrition during and

177 after hospitalization are similar for all OIs. One exception is that patients with signs of severe

178 acute kidney injury had worse outcomes than patient with other conditions with 23 (57%)

179 p=0.002 dying during hospitalization, and 7 (50%) p=0.650 being LTFU or dead during the 6

180 months of follow up.”

Generally speaking it is best to give some less details to allow a better understanding

When you discuss the viability of ARV first line (NNRTI) please report the WHO guidelines to make understandable to the reader whether this is in agreement or not with the mainstream on this issue.

6. PLOS authors have the option to publish the peer review history of their article (what does this mean?). If published, this will include your full peer review and any attached files.

Reviewer #1: **Yes: **Getandale Zeleke Negera

Reviewer #2: No

---

## [Author Response · Author response to Decision Letter 0]

23 Jul 2021

please find the information regarding revierwers suggestions as well as changes to the manuscript in the rebuttal letter and the track changes version of the manuscript

---

## [Decision Letter · Decision Letter 1]

16 May 2022

PONE-D-20-31177R1Poor outcomes among critically ill HIV-positive patients at hospital discharge and post-discharge in Guinea, Conakry: A retrospective cohort studyPLOS ONE

Dear Dr. Albus,

Thank you for submitting your manuscript to PLOS ONE. After careful consideration, we feel that it has merit but does not fully meet PLOS ONE’s publication criteria as it currently stands. Therefore, we invite you to submit a revised version of the manuscript that addresses the points raised during the review process.

We look forward to receiving your revised manuscript.

Kind regards,

Mohammad Bellal Hossain

Academic Editor

PLOS ONE

Journal Requirements:

Reviewers' comments:

Reviewer's Responses to Questions

**Comments to the Author**

1. If the authors have adequately addressed your comments raised in a previous round of review and you feel that this manuscript is now acceptable for publication, you may indicate that here to bypass the “Comments to the Author” section, enter your conflict of interest statement in the “Confidential to Editor” section, and submit your "Accept" recommendation.

Reviewer #1: All comments have been addressed

Reviewer #2: (No Response)

2. Is the manuscript technically sound, and do the data support the conclusions?

Reviewer #1: Yes

Reviewer #2: Yes

3. Has the statistical analysis been performed appropriately and rigorously? 

Reviewer #1: No

Reviewer #2: Yes

4. Have the authors made all data underlying the findings in their manuscript fully available?

Reviewer #1: Yes

Reviewer #2: Yes

5. Is the manuscript presented in an intelligible fashion and written in standard English?

Reviewer #1: Yes

Reviewer #2: Yes

6. Review Comments to the Author

Reviewer #1: Dear editor,

the authors have addressed all of my concerns. I hope this piece of article would contribute to filling the gap in the field, particularly in Sub-saharan Africa, the region that carries the highest burden of HIV infection. Thank you.

Reviewer #2: The article is interesting and well written.

Data about HIV patient care in WA are scarce and needed.

Some minor review are suggested:

- is it possible to list the health units where the patients were followed up after discharge?

- is it possible to include reason for admission? is it not clear why the patients were admitted at hospital, could it be related with baseline characteristics and/or outcome? This could be a point to discuss when you compare your results with other study... were their patients admitted for the same reasons than yours? This is also an important information in the epidemiological point of view. Your conclusion about HIV severity in low-resource settings is valid, but the results of your study are focusing only on patients admitted at hospital (for what reason?), so has limited possibility to be extended to the general population. Nonetheless, it is an important factor to highlight on a public health point of view (i.e. knowing what burden of HIV diseases the hospitals have to face in WA). Consider adding some discussion about that in the discussion section.

- why malnutrition was defined with MUAC and not with BMI? Is BMI data available? Consider move the definition of malnutrition in the methods section and not in the results.

7. PLOS authors have the option to publish the peer review history of their article (what does this mean?). If published, this will include your full peer review and any attached files.

Reviewer #1: **Yes: **Getandale Zeleke Negera (PhD student)

Reviewer #2: No

---

## [Author Response · Author response to Decision Letter 1]

18 Jul 2022

please find all necessary comments in the current rebuttal letter

---

## [Decision Letter · Decision Letter 2]

2 Sep 2022

PONE-D-20-31177R2Poor outcomes among critically ill HIV-positive patients at hospital discharge and post-discharge in Guinea, Conakry: A retrospective cohort studyPLOS ONE

Dear Dr. Albus,

Thank you for submitting your manuscript to PLOS ONE. After careful consideration, we feel that it has merit but does not fully meet PLOS ONE’s publication criteria as it currently stands. Therefore, we invite you to submit a revised version of the manuscript that addresses the points raised during the review process.  Please submit your revised manuscript by Oct 17 2022 11:59PM. If you will need more time than this to complete your revisions, please reply to this message or contact the journal office at plosone@plos.org. Please include the following items when submitting your revised manuscript:A rebuttal letter that responds to each point raised by the academic editor and reviewer(s). You should upload this letter as a separate file labeled 'Response to Reviewers'.A marked-up copy of your manuscript that highlights changes made to the original version. You should upload this as a separate file labeled 'Revised Manuscript with Track Changes'.An unmarked version of your revised paper without tracked changes. You should upload this as a separate file labeled 'Manuscript'.If applicable, we recommend that you deposit your laboratory protocols in protocols.io to enhance the reproducibility of your results. Protocols.io assigns your protocol its own identifier (DOI) so that it can be cited independently in the future. For instructions see: https://journals.plos.org/plosone/s/submission-guidelines#loc-laboratory-protocols. Additionally, PLOS ONE offers an option for publishing peer-reviewed Lab Protocol articles, which describe protocols hosted on protocols.io. Read more information on sharing protocols at https://plos.org/protocols?utm_medium=editorial-email&utm_source=authorletters&utm_campaign=protocols.

We look forward to receiving your revised manuscript.

Kind regards,

Mohammad Bellal Hossain

Academic Editor

PLOS ONE

Journal Requirements:

Reviewers' comments:

Reviewer's Responses to Questions

**Comments to the Author**

1. If the authors have adequately addressed your comments raised in a previous round of review and you feel that this manuscript is now acceptable for publication, you may indicate that here to bypass the “Comments to the Author” section, enter your conflict of interest statement in the “Confidential to Editor” section, and submit your "Accept" recommendation.

Reviewer #1: All comments have been addressed

Reviewer #2: All comments have been addressed

2. Is the manuscript technically sound, and do the data support the conclusions?

Reviewer #1: Yes

Reviewer #2: Partly

3. Has the statistical analysis been performed appropriately and rigorously? 

Reviewer #1: I Don't Know

Reviewer #2: Yes

4. Have the authors made all data underlying the findings in their manuscript fully available?

Reviewer #1: Yes

Reviewer #2: Yes

5. Is the manuscript presented in an intelligible fashion and written in standard English?

Reviewer #1: Yes

Reviewer #2: Yes

6. Review Comments to the Author

Reviewer #1: I really want to appreciate the authors for taking the initiative to conduct such a terrific article in one of the countries with a high HIV burden.

Reviewer #2: The paper is interesting and provides important results about HIV care in west Africa.

In my opinion the results section should be improved.

The tables should include a column for retained patients. For instance table 1 should have: Total, Died, Alive. So that you can compare dead with alive patients. The same for the other tables.

The narrative description of data reported in figures (in particular in the section "Baseline demographic and clinical characteristics") is not so clear and should be improved. For instance, it is not clear what do you refer to when discussing about "both cohorts".

When improved these sections, the paper can be considered for publication.

7. PLOS authors have the option to publish the peer review history of their article (what does this mean?). If published, this will include your full peer review and any attached files.

Reviewer #1: **Yes: **Getandale Zeleke Negera

Reviewer #2: No

---

## [Author Response · Author response to Decision Letter 2]

23 Sep 2022

Reviewer 2 found the results section of the paper needing of improvement:

First reviewer 2 suggested changing table 1 and 2 so they would show both dead/LTFU as well as patients still alive. We appreciate the comment and changed both table 1 and 2 to the suggestion of reviewer 2. Please note however that as a consequence we struggled to fit the tables on A4 portrait page format. Please feel free to change the formatting accordingly to comply with journal requirements before publishing.

Second, reviewer 2 found the narrative description of the section „baseline demographic and clinical characteristics“ unclear. We appreciate his comments and improved the wording, including more information on the two cohorts that were included in the analysis. For details, we would like to refer reviewers to the uploaded document with track changes.

---

## [Decision Letter · Decision Letter 3]

24 Jan 2023

Poor outcomes among critically ill HIV-positive patients at hospital discharge and post-discharge in Guinea, Conakry: A retrospective cohort study

PONE-D-20-31177R3

Dear Dr. Albus,

We’re pleased to inform you that your manuscript has been judged scientifically suitable for publication and will be formally accepted for publication once it meets all outstanding technical requirements.

Kind regards,

Owen Ngalamika

Academic Editor

PLOS ONE

Additional Editor Comments (optional):

Reviewers' comments:

Reviewer's Responses to Questions

**Comments to the Author**

1. If the authors have adequately addressed your comments raised in a previous round of review and you feel that this manuscript is now acceptable for publication, you may indicate that here to bypass the “Comments to the Author” section, enter your conflict of interest statement in the “Confidential to Editor” section, and submit your "Accept" recommendation.

Reviewer #1: All comments have been addressed

Reviewer #2: All comments have been addressed

2. Is the manuscript technically sound, and do the data support the conclusions?

Reviewer #1: Yes

Reviewer #2: Yes

3. Has the statistical analysis been performed appropriately and rigorously? 

Reviewer #1: Yes

Reviewer #2: Yes

4. Have the authors made all data underlying the findings in their manuscript fully available?

Reviewer #1: Yes

Reviewer #2: Yes

5. Is the manuscript presented in an intelligible fashion and written in standard English?

Reviewer #1: Yes

Reviewer #2: Yes

6. Review Comments to the Author

Reviewer #1: The authors significantly improved the manuscript compared with the initial submission and I strongly recommend the editor to consider the article for publication. They have also fully addressed my review comments.

Reviewer #2: The paper is now improved as all the comments have been addressed. The authors provide interesting data about HIV care in west Africa. The paper is now valuable for publication.

7. PLOS authors have the option to publish the peer review history of their article (what does this mean?). If published, this will include your full peer review and any attached files.

Reviewer #1: **Yes: **Getandale Zeleke Negera

Reviewer #2: No

---

## [Editor Report · Acceptance letter]

2 Mar 2023

PONE-D-20-31177R3 

Poor outcomes among critically ill HIV-positive patients at hospital discharge and post-discharge in Guinea, Conakry: A retrospective cohort study 

Dear Dr. Albus:

I'm pleased to inform you that your manuscript has been deemed suitable for publication in PLOS ONE. Congratulations! Your manuscript is now with our production department. 

Kind regards, 

on behalf of

Dr. Owen Ngalamika 

Academic Editor

PLOS ONE